# Anti-Inflammatory Effects of Analogues of *N*-Acyl Homoserine Lactones on Eukaryotic Cells

**DOI:** 10.3390/ijms21249448

**Published:** 2020-12-11

**Authors:** Agathe Peyrottes, Garance Coquant, Loïc Brot, Dominique Rainteau, Philippe Seksik, Jean-Pierre Grill, Jean-Maurice Mallet

**Affiliations:** 1Laboratoire des Biomolécules (LBM), Département de chimie, École Normale Supérieure, PSL University, Sorbonne Université, CNRS, 75005 Paris, France; agathepeyrottes@gmail.com (A.P.); Jean-Maurice.Mallet@ens.fr (J.-M.M.); 2INSERM, Centre de Recherche Saint-Antoine, APHP, Hôpital Saint-Antoine, Microbiote Intestin et Inflammation, Sorbonne Université, 75005 Paris, France; garance.coquant@gmail.com (G.C.); loic.brot@sorbonne-universite.fr (L.B.); dominique.rainteau@sorbonne-universite.fr (D.R.); jean-pierre.grill@upmc.fr (J.-P.G.); 3Service de Gastroentérologie et Nutrition, Hôpital Saint-Antoine, APHP, 75012 Paris, France

**Keywords:** AHL, inflammatory bowel disease, gut microbiota

## Abstract

Background: Since acyl-homoserine lactone (AHL) profiling has been described in the gut of healthy subjects and patients with inflammatory bowel disease (IBD), the potential effects of these molecules on host cells have raised interest in the medical community. In particular, natural AHLs such as the 3-oxo-C12-HSL exhibit anti-inflammatory properties. Our study aimed at finding stable 3-oxo-C12-HSL-derived analogues with improved anti-inflammatory effects on epithelial and immune cells. Methods: We first studied the stability and biological properties of the natural 3-oxo-C12-HSL on eukaryotic cells and a bacterial reporter strain. We then constructed and screened a library of 22 AHL-derived molecules. Anti-inflammatory effects were assessed by cytokine release in an epithelial cell model, Caco-2, and a murine macrophage cell line, RAW264.7, (respectively, IL-8 and IL-6) upon exposure to the molecule and after appropriate stimulation (respectively, TNF-α 50 ng/mL and IFN-γ 50 ng/mL, and LPS 10 ng/mL and IFN-γ 20 U/mL). Results: We found two molecules of interest with amplified anti-inflammatory effects on mammalian cells without bacterial-activating properties in the reporter strain. The molecules furthermore showed improved stability in biological medium compared to the native 3-oxo-C12-HSL. Conclusions: We provide new bio-inspired AHL analogues with strong anti-inflammatory properties that will need further study from a therapeutic perspective.

## 1. Introduction

The gastrointestinal tract is a complex milieu in which host epithelial and immune cells coexist with a wide array of microorganisms, namely gut microbiota and food antigens. The gut microbiota is composed of trillions of bacteria (10^13^ bacteria) that collectively play major roles in human physiology, being notably involved in the intestinal barrier and immune system maturation [1]. Gut microbiota metabolites such as short-chain fatty acids, bile acids, and tryptophan metabolites are the main known drivers of the impact of gut microbiota on hosts [2]. Manipulating gut microbiota, and especially microbiota derived metabolites, to influence host physiology remains therefore a major challenge in an increasing number of human diseases. This approach seems perfectly suited in inflammatory bowel diseases (IBD) where dysbiosis (imbalance in gut microbiota) and inflammation are seen [3]. A largely overlooked component of gut microbiota is that of diffusible signal molecules, which are known to modulate the physiological response in human cells. A particular class of these signaling compounds is represented by bacterial quorum sensing (QS) molecules and particularly *N*-acyl-homoserine lactones (AHLs).

Because eukaryotic and prokaryotic cells have co-existed and co-evolved together, they have adapted to the signal molecules secreted by one another. AHLs are known to influence gene expression and certain behaviors in eukaryotes, a phenomenon called interkingdom signaling [4,5,6]. To date, most studies on the effects of AHLs on mammal cells have focused on the well-known molecules 3-oxo-C12-HSL (1) and C4-HSL from the pathogenic bacteria *Pseudomonas aeruginosa*, and 3-oxo-C6-HSL from *Vibrio fischeri* for medical and historic reasons [7,8,9,10]. AHLs have been described to enter human cells, where they can exert various effects ranging from apoptosis induction to immune and inflammation modulation [7,8,11,12,13]. Not only can AHLs recruit specific cell types, but they also modulate the cytokine secretion in vitro at both the RNA and protein levels [14]. AHL effects have been described to date on very diverse cell types, from breast cancer to immune cells, but also lung epithelium, and more recently on intestinal epithelial cells [14,15,16].

Indeed, QS driven by AHLs has been described in many bacterial ecosystems, but it has not been well studied in the human intestinal microbiota. We published the first detection by LC-MS/MS of AHLs from the human gut, and profiled AHLs distribution over a cohort of healthy individuals and IBD patients [17]. This led to the identification of the most conserved, abundant, and never-described AHL associated to the healthy human gut: It is a doubly unsaturated AHL of formula 3-oxo-C12:2-HSL (2) [17]. Unsaturated AHLs are not commonly found in bacteria, which favor saturated molecules [18]. To date, only the marine *Roseobacter* clade is a well-known producer of mono- and di-unsaturated AHLs, with carbon chains ranging from 10 to 18 atoms [19,20,21]. However, none of these molecules carry a ketone [19,20,21]. Hence, the discovery of new active forms of AHLs questions the tight relationship between their chemical structure and their bioactivity in both mammals and bacteria. In fact, rates of 3-oxo-C12:2-HSL in feces of individuals negatively correlates with IBD status: healthy controls and patients in remission exhibited high levels of the molecule, while patients with active disease showed no or little amounts of this molecule [17]. This led to the hypothesis of an anti-inflammatory effect of the 3-oxo-C12:2-HSL. We were able to show a modest anti-inflammatory effect of 3-oxo-C12:2-HSL on epithelial cells [17].

In order to enhance this effect, we searched for more powerful and more stable molecules. Using the natural AHL 3-oxo-C12-HSL as a template, we here report the development of several AHL analogues. The newly designed molecules were then screened for bioactivity on both mammalian cell lines and bacterial reporter strains.

## 2. Results

### 2.1. The Natural AHL 3-Oxo-C12-HSL Exerts Modest Anti-Inflammatory Effects on Gut Barrier Eukaryotic Cells

#### 2.1.1. AHL 3-Oxo-C12-HSL Exerts Anti-Inflammatory Effects on Epithelial Cells

Non stimulated Caco-2/TC7 cells exposed with increasing doses of 3-oxo-C12-HSL did not show significant change in interleukine-8 (IL-8) secretion. When epithelial cells were stimulated with IL-1β, we observed a significant decrease in IL-8 secretion in the presence of 5 μM 3-oxo-C12-HSL, compared to that of the control DMSO (Figure 1a). There was a bell-shaped activity curve, with significant decreases of IL-8 secretion at low concentrations (range 1–10 µM), while the effects were abolished at higher concentrations (over 50 µM). The maximum magnitude was reached with a dose of 5 µM, where a reduction of 27.6% of secreted IL-8 was observed compared to that of control. Of note, no toxicity of 3-oxo-C12-HSL was observed on the epithelial cell line Caco-2/TC7 (Appendix A).

#### 2.1.2. AHL 3-Oxo-C12-HSL Anti-Inflammatory Effects on Macrophage Cell Line are Milder

Non-stimulated RAW264.7 macrophages challenged to increasing doses of 3-oxo-C12-HSL did not show changes in IL-6 secretion compared to that of control DMSO. In murine macrophage cell lines stimulated with lipopolysaccharide (LPS) and interferon-γ (IFNγ), we observed a decrease in IL-6 secretion when adding 3-oxo-C12-HSL at 15 μM as compared to that of the control (Figure 1b). Yet, the reduction was only 17% compared to that of control and did not reach statistical significance. Moreover, no toxicity of 3-oxo-C12-HSL was found between 1 and 50 µM (Appendix A).

### 2.2. The Natural AHL 3-Oxo-C12-HSL but Not C4-HSL Activates LasR

#### AHL Are Natural Bacterial Modulators and 3-Oxo-C12-HSL Can Activate the LasR Receptor

3-oxo-C12-HSL was tested on the bacterial reporter strain E. coli pSB1075 to assess its interactions with the *P. aeruginosa* AHL receptor, LasR, and the resulting activation. As expected, the 3-oxo-C12-HSL elicited a strong activation of LasR, with a read EC_50_ of 9 nM (Figure 1c). In contrast, the short-chain AHL from *P. aeruginosa*, C4-HSL, showed no induction of luminescence and was then used as a negative control (Figure 1c).

### 2.3. AHL 3-Oxo-C12-HSL Is an Unstable Molecule in Biological Media

#### 2.3.1. The AHL Instability Results from Its Chemical Characteristics

Degradation at biological pH affords by-products (3) and (4), respectively, called tetramic acid and open-form AHL or 3-oxo-C12-HS (Figure 2a).

The chemical composition of a solution of 3-oxo-C12-HSL in non-conditioned cell-free DMEM was studied at room temperature over 24 h. AHL decay gave the expected by-products (3) and (4), and kinetic constants were interpolated using standard curve solutions (Figure 2b). The 3-oxo-C12-HSL degradation follows an apparent first-order mechanism, with apparent constants k_app_ and t_1/2_, respectively, 0.2 h^−1^ and 3.8 h. Overall, the AHL hydrolysis in cell medium was observed within hours, with product (4) largely predominant over product (3) (91:9 ratio).

#### 2.3.2. Biological Study of Tetramic Acid (3)

The degradation product tetramic acid (3) from 3-oxo-C12-HSL was synthesized and its bioactivity studied on both the Caco-2/TC7 and RAW264.7 cell lines and the bacterial reporter. Compared to control, the molecule showed no effect on cell lines but significant cytotoxicity (Appendix A). In the AHL receptor reporter strain, the molecule acted as a weak agonist of LasR, with an apparent EC_50_ 300-fold greater than that of the parent AHL 1 (2.7 µM vs. 9 nM) (Figure 3). Overall, our results suggest tetramic acid (3) cannot be accountable for the observed modulatory effects of 3-oxo-C12-HSL.

### 2.4. Intracellular Traffic of 3-Oxo-C12-HSL

The entry and distribution of 3-oxo-C12-HSL over time in Caco-2/TC7 cells was examined, in the presence and absence of 2-hydroxyquinoline (2-HQ). 2-HQ inhibits enzymatic PON-mediated hydrolysis of the AHL lactone. The distribution of the inhibitor is predominantly extracellular (Appendix A). In the presence of 2-HQ, 50% of the AHLs entered the cells within 5 min, and approx. 90% were found in cell lysates after 1 h. Without 2-HQ, only 18% of the AHLs were intracellular after 5 min, and only 65% of the AHLs were internalized over experiment time (Figure 3a). From these observations we hypothesized that inhibition of PON hydrolysis in the extracellular environment promotes the maintenance of AHL integrity, and thereby its penetration into cells.

While studying the proportional repartition of intact 3-oxo-C12-HSL (1) and degraded 3-oxo-C12-HS (4), it appeared that (1) rapidly hydrolysed into its open homoserine (Figure 3b). By investigating the ratio between molecules (1) and (4) in total cell lysates we observed that the open-form (4) became prevalent within minutes after cell penetration, and was the major form over longer timespans. The absence of 2-HQ retarded the appearance of 3-oxo-C12-HSL’s peak concentration from 5 min to 20 min, otherwise no significant difference was observed with or without inhibitor. Overall, the open form dominated in cells, representing between 60 and 95% of the total amount found in cell lysates.

The 3-oxo-C12-HSL limited stability in biological media, and its activating properties on bacteria are ill-suited for direct use as an ecological drug, although the AHL evidently possesses mild anti-inflammatory properties. However, a library of analogues was designed in order to take advantage of the latter anti-inflammatory properties. It allowed for the identification of chemical structures in our cell models, without exhibition of bacterial activation and stability issues.

### 2.5. Structure-Activity Relationship in AHL and Identification of Two Hit Compounds

#### 2.5.1. Construction of a Library of AHL Analogues

Using 3-oxo-C12-HSL (1) as a framework, chemical characteristics of the AHL molecular class were challenged for their impact on biological activity. Chemical modifications were performed on the acyl chain, the C3-ketone and the lactone headgroup, to build a total library of 22 compounds (Table 1).

The synthetic strategy was adapted from published literature (Appendix A), and enhanced with additional steps (Appendix A), to afford increased yields, easier purifications, and improved synthetic flexibility [22,23,24,25,26,27]. The total synthesis of natural AHL and bio-inspired analogues started with the coupling of an appropriate carboxylic acid to Meldrum’s acid. The resulting intermediate was refluxed in methanol to yield the corresponding carboxylic acid methyl ester. The ketone was protected with ethylene glycol, and the terminal ester hydrolyzed to the free carboxylic acid. This acid was coupled to the appropriate headgroup, and finally the ketone deprotection afforded the target compound. Meanwhile, the synthesis of non-3-oxo AHLs was straightforward and required only one step: coupling of an appropriate carboxylic acid with the headgroup (Appendix A). The resulting library contained natural AHLs, analogues, and other closely related molecules, spanning across a variety of acyl chain lengths, headgroups, and C3 substitution patterns.

#### 2.5.2. Compared Results of the Two New Molecules on Mammalian Cell Lines versus the Parent AHL

Analogue (*S*,*S*)-3-oxo-C12-ACH (30): Although, this analogue showed a similar level of reduction of IL-8 secretion in IL-1β stimulated Caco-2/TC7 cells as that of 3-oxo-C12-HSL (1) (Figure 4a), its effect was stronger on RAW264.7 macrophages. The (*S*,*S*)-isomer analogue revealed a 52% reduction of IL-6 at 50 μM (Figure 4b), without cytotoxicity (Appendix A).

Analogue 3-oxo-C12-2-amino-4-chlorophenol (35): Compared to 3-oxo-C12-HSL (1), this analogue showed a 50% reduction of IL-8 secretion by stimulated Caco-2/TC7 epithelial intestinal cells with a clear dose-dependent effect (Figure 5a). Moreover, we observed a 60% reduction in IL-6 secretion by stimulated RAW264.7 at 15–50 µM. There was no cytotoxicity on both studied cell lines induced by analogue (35) (Appendix A). This analogue appears the most active of our library (Figure 5b).

#### 2.5.3. The (*S*,*S*)-3-Oxo-C12-ACH (30) Activated LasR While the 3-Oxo-C12-2-amino-4-chlorophenol (35) Did Not

Analogue (*S*,*S*)-3-oxo-C12-ACH (30) is an agonist of the bacterial receptor with EC_50_ of 170 nM (Figure 4c. In contrast, molecule (35) showed an inability to induce luminescence when incubated with the biosensor (Figure 5c). After demonstrating that the two analogues were non-toxic to the bacteria, the ACP analogue (3-oxo-C12-2,4-aminochlorophenol) was tested in an inhibition assay (Figure 5d). It was concluded that molecule (35) was a weak LasR antagonist, with an IC_50_ of 28.2 µM.

## 3. Discussion

Natural 3-oxo-C12-HSL exerts modest anti-inflammatory effects on epithelial and macrophage cell lines but remains unstable. Our study indicates that AHL analogues with stronger biological effects can be found from a synthetic library derived from 3-oxo-C12-HSL (1). This strategy allowed us to find out a stable analogue, 3-oxo-C12-2-amino-4-chlorophenol (35), exerting a strong anti-inflammatory effect on both cell lines without activating the bacterial AHL receptor LasR. These properties make it a good candidate to be a new anti-inflammatory molecule to prevent intestinal inflammation.

### 3.1. Stability and Concentration

As signal molecules, it is not unexpected that AHLs degrade rapidly after secretion. Indeed, AHL-sensing allows bacteria to detect other members of the colony. Hence it is important that AHLs get degraded to ensure their concentration closely reflects the temporal dynamics of the bacterial population. Human niches represent bio-diverse environments where AHLs could play a signaling role. In the pulmonary ecosystem, Losa et al. demonstrated parallel evolutions between 3-oxo-C12-HSL concentration and *P. aeruginosa* loads [28]. In the gut ecosystem, our group previously showed that 3-oxo-C12:2-HSL correlates to normobiosis, higher counts of Firmicutes, and lower counts of *E. coli* [17]. However, the 3-oxo-C12-HSL is unstable at biological pH due to two main degrading phenomena: a spontaneous, irreversible, and non-enzymatic rearrangement according to a Claisen-like mechanism to produce tetramic acid (3) and the hydrolysis of its lactone ring to produce the open-form (4). Our results suggest that these by-products do not have anti-inflammatory properties. Moreover, we have shown, in accordance with the literature that 3-oxo-C12-HSL can enter cells within minutes [29,30]. This compromises pH-dependent hydrolysis in the extracellular compartment and rises the problem of AHL enzymatic degradation.

Our experiments described a short half-life for 3-oxo-C12-HSL incubated at 37 °C in the presence of Caco-2/TC7 cells. This result is concordant with findings from Losa et al. on polarized bronchial epithelial cells [28]. Indeed, eukaryotic cells secrete degrading enzymes that catalyze lactonolysis reactions to produce open-ring AHLs, called lactonases. In particular, lactone-hydrolyzing paraoxonases (PON) are expressed in most human tissues, including the intestine [31], and are conserved in post-confluency, differentiated, and polarized Caco-2 cells [32,33,34,35]. Our results show that using a competitive PON inhibitor in the extracellular compartment increases the half-life of undegraded AHLs in cell supernatant and improves their capacity to penetrate the intracellular compartment. Inside cells, the molecule is not protected by the PON inhibitor anymore, and rapidly hydrolyzes into 3-oxo-C12-HS. Kravchenko et al. made similar findings on macrophages [36]. Their results were later confirmed by Horke et al. on several PON2-producing cell lines [37]. We believe that PON are required intracellularly for AHL activity, while their extracellular presence will prevent AHL cell penetration and thus decrease the observed effects. This is supported by literature demonstrating that the 3-oxo-C12-HSL cellular effects were reversed after incubation with PON inhibitor TQ416, suggesting that AHL requires PON-mediated intracellular hydrolysis to perform its function [37,38,39]. Our hypothesis does not refute the statement that the structural integrity of the lactone ring is required to mediate its effects on mammalian cells [27,36]; but rather we believe that lactone is required to mediate AHL penetration into cells, like a pro-drug.

It seems quite important to examine the dosage of AHL employed in our experiments and its biological relevance. AHLs have been identified in vivo in an array of fluids, from human sputum to bacterial biofilms, and now in feces as reported by our team [17]. “Indirect” sampling, where samples are not taken in the immediate cellular surroundings, shows very low AHL concentrations. It is, however, probable that this method may only provide insights of the actual AHL levels or may undergo technical bias. Thus, concentrations ranging from 1 to 20 nM were detected in the sputum of patients with cystic fibrosis [40], while 0.2 to 2 nM/g AHL were found in fecal samples from IBD patients. In contrast, very high concentrations of AHL are measured in *P. aeruginosa* biofilms: levels as high as 300–600 µM have been reported by Charlton et al., and biofilms have been identified in the lungs of infected cystic fibrosis patients [41]. Our work aimed at exploring AHL’s effects on cells lines in the context of gut inflammation driven by local dysbiotic microbiota. Planktonic *P. aeruginosa* secrete AHLs at low micromolar concentrations [42], similar to the concentrations detected in a murine model of *P. aeruginosa* acute lung infection (1–20 µM). Hence, we thought it was relevant to broaden this concentration range to 1–100 µM, which could be the AHL dose range that intestinal cells from an IBD patient would encounter in a non-infected but inflammatory and dysbiotic state.

### 3.2. Choice of Chemical Modifications

Requirements for full AHL activity on mammalian cell lines are traditionally described as follows: integrity of the lactone head, an acyl chain from 10–14 carbons, and an oxo or hydroxyl substitution at carbon-3 [27,36,43]. Our chemical modifications aimed at challenging each of these requirements and identifying analogues of 3-oxo-C12-HSL with valuable activity.

Eight molecules (2, 9, 26, 27, 42, 43, and 44) were chosen to investigate the influence of ketone substitution, acyl chain length, substitution, and saturation. Fifteen analogues with modified headgroups but that retained the other native patterns (12-carbon-long acyl chain and 3-oxo substitution) were synthesized to study the involvement of the lactone headgroup in activity. The headgroups were selected according to four parameters: the stereochemistry of the α-carbon to the lactone ring, the influence of electron-donating atoms in the ring, the ring size, and the capacity to be hydrolyzed.

The stereochemistry of the lactone is crucial in natural AHL, where only the (*L*)-enantiomer is active. Chhabra et al. have in particular demonstrated that the immunomodulatory activity of AHLs depended critically on their (*L*)-conformation [27]. Where possible, the (*S*)-stereochemistry was conserved and the reverse enantiomer was synthesized for comparison, while some headgroups were voluntarily selected for their planar structure. All headgroups but one included electron-donating atoms, whose lone pairs promote H-bonds formation and non-covalent interactions of the molecule in its active site. Most of the synthetic compounds cannot be hydrolyzed because they are resistant to lactonolysis. Three hydrolysable molecules were included: the (*R*)-lactone and thiolactone rings remain sensitive to hydrolysis.

### 3.3. Structure–Activity Relationship in AHL and Analogues

Our molecular library highlighted conservative and deleterious modifications to AHL cellular activity. Overall, we confirmed the need for the acyl chain and carbon-3 substitution. Maximum activity is achieved with 12–14 carbon-long chains, and carbon substitution with linear heteroatoms has moderate effects. Addition of large groups on C3 abolishes activity, while the complete removal of substituent weakens it. Results on AHL ring substitution have widened the spectrum of possibilities, until now believed to be restricted to homoserine groups only. Headgroups that retained activity include the homocysteine lactone rings, amino-cyclohexanol, and amino-chlorophenol. Surprisingly, the thiolactone group, closest to native HSL in terms of structure, retained good biological activity but was not the most active. Both lead candidates (*S*,*S*)-3-oxo-C12-ACH and 3-oxo-C12-2-amino-4-chlorophenol exhibit common features: they are 6-membered rings with hydroxyl group in the *ortho* position. The major difference is the nature of those rings, which impacts their 3D structures and spatial interactions. Aromatic rings are less flexible than saturated rings and cannot accommodate to best fit their receptor, but their locked conformation can confer energetic advantages. This assumption should, however, be verified by proper docking and partner investigation.

Bacterial results are in accordance with the published literature. Modest modifications of the alkyl chain have important impacts on the molecule capacity to bind the LasR receptor. Only tail length ranging from 10–14 carbons retains activity in the nanomolar range, including analogues with a heteroatom. Besides length, alkyl chain flexibility seems required to retain activity, as confirmed in the literature [44,45]. Smith et al. pioneered the investigation of AHL analogues with aminocyclohexanol headgroups. The group described the (*S*,*S*)-isomer (30) as a LasR agonist with inducing properties similar to that of native 3-oxo-C12-HSL [46]. The authors hypothesized that the hydroxyl group of amino-cyclohexanol appropriately mimics the carbonyl group of native AHL to participate in H-bonds, while the cyclohexane ring mimics the ring structure in hydrophobic interactions.

Aromatic AHL analogues have been described as LasR antagonists by several groups [25,46,47,48,49]. Observing that both (35) and a 2-aminophenol AHL analogue exhibited similar LasR antagonisms, Smith et al. assumed the pattern of adjacent hydroxyl and amino groups was required for activity in aromatic rings, like in aminocyclohexanol analogues. The unsaturated carbon bonds would then create a difference in behavior between agonism and antagonism.

It is important to note that 3-oxo-C12-2-amino-4-chlorophenol (35) best meets the criteria we defined for lead candidate. On top of demonstrating improved activity on mammalian cell assays and being PON-resistant, this analogue does not positively modulate the bacteria LasR receptor. The molecule is furthermore not toxic for bacterial populations. Bactericidal properties could indeed be deleterious in dysbiotic and immuno-compromised individuals, as they would make ecological niches available for colonization by pathogenic bacterial communities. In contrast, analogue (*S*,*S*)-3-oxo-C12-ACH (30) exhibits promising activity on mammalian models, but its LasR agonistic behavior might prevent it from future in vivo experimentation.

### 3.4. Future Work

Form our results, one can imagine testing molecules of interest, such as the 3-oxo-C12-2-amino-4-chlorophenol, in vivo. Indeed, our study paved the way to understanding the relationships between AHL-derived molecules in an environment as complex as the intestine. The stability of the molecule, the absence of recognition by AHL receptors, and its anti-inflammatory effect on murine macrophages indicate the next steps of our study. In this setting, a murine model of colitis appears a good model to evaluate the impact of (35) on gut inflammation without modifying gut microbiota. Another implementation of our work could be to synthesize labelled AHLs of interest to perform molecular imaging and functional assays. Labelling could be done at multiple positions. The substitution should preserve the whole structure and be “transparent”. We are currently thinking of short PEG groups attached to a biotin or a fluorescein in position C12, or on the 3-keto group. The resulting fluorescent analogue (35) could then be used for cell and tissue imaging, and to localize and follow the fate of the compound in epithelial and immune cells. Using the same method, a biotinylated molecule could help identify molecular partners in vivo. For instance, AHL-coated beads can be used to extract relevant ligand proteins from human intestinal epithelial Caco-2/TC7 cells. This technique was already proven successful in finding eukaryotic ligands (T2R38 and IQGAP1) of the 3-oxo-C12 AHL [50,51].

To conclude, gut microbiota mutually interacts with co-evolved host epithelial and immune cells in a beneficial reciprocal relationship. QS-signaling of bacteria probably contribute substantially to establishing symbiotic interactions in some cross-kingdom interactive dynamics. Searching for QS-derived molecules to control host immune responses appears to be a completely novel and original approach from a medical perspective. Our study opens this new field of research showing that more stable and efficient anti-inflammatory molecules derived from AHLs can be found. Moreover, development of such an approach could help in understanding the mechanisms of action and degradation of these diffusible signaling molecules that impact human gut physiology.

## 4. Materials and Methods

### 4.1. Reagents

2-hydroxyquinoline, 3-oxo-C12-HSL, and sterile DMSO were bought from Sigma. All chemical reagents were purchased from commercial suppliers Sigma-Aldrich (Saint-Louis, MO, USA) and TCI Chemicals (Paris, France), and used without further purification.

### 4.2. Cell Culture Methods

Caco-2/TC7 cells were seeded at 10^5^ cells/well in 6-well plastic culture plates and maintained in high glucose medium (DMEM GlutaMAX 4.5 g/L glucose, ThermoFisher Scientific Waltham, MA, USA) supplemented with 20% heat-inactivated fetal calf serum (FCS, GE Healthcare, Life Science, Chicago, IL, USA), 1% non-essential amino acids NEAA, and 1% penicillin-streptomycin (ThermoFisher Scientific, Waltham, MA, USA). The cells were cultured at 37 °C in a 10% CO_2_/air atmosphere. The media were changed every day. Confluency was reached on day 6. On day 17, the cells were serum-starved, and used for stimulation on day 18.

RAW264.7 murine macrophage cells (ATCC, Manassas, VA, USA) were cultured in DMEM supplemented with 10% heat-inactivated fetal calf serum and 1% 200 mM L-glutamine (ThermoFisher Scientific, Waltham, MA, USA). The macrophages were seeded at 40,000 cells/well in 24-well plates, upon reaching 80–90% confluence after 4-day culture. The cells were maintained at 37 °C with 5% CO_2_/air atmosphere and the media were changed every other day.

### 4.3. Cell Stimulation Protocols

For stimulation, Caco-2/TC7 cell medium was serum-free and supplemented with 100 µM 2-HQ (Sigma, Saint-Louis, Missouri, USA). The cells were incubated for 18 h at 37 °C with stimulation medium containing 0.1% DMSO or tested compounds, with or without a pro-inflammatory cytokine (IL-1ß at 25 ng/mL). After 18 h, the supernatants were collected, and stored at −80 °C before IL-8 quantification by ELISA (Duoset Human CXCL8/IL-8, R&D Systems, Minneapolis, MN, USA), according to manufacturer’s instructions. Cells were washed with PBS 1X and lysed in PBS 1X containing 1% triton X-100. Cells were harvested by scraping and stored at −80 °C before protein quantification by BCA assay (Interchim, Montluçon, France).

RAW264.7 cells were incubated for 6 h at 37 °C with stimulation medium containing 0.1% DMSO or tested compounds, with or without LPS (10 ng/mL, Sigma, Saint-Louis, Missouri, USA) and IFN-γ (20 U/mL, R&D systems, Minneapolis, MN, USA) to establish inflammation. After 6 h, the supernatants were collected, and stored at −80 °C before IL-6 quantification by ELISA (BD OptEIA Mouse Il-6 ELISA Set kit BD Biosciences, Eysins, Vaud. Switzerland), according to manufacturer’s instructions. Cells were harvested by scraping and stored at −80 °C before protein quantification by BCA assay (Interchim, Montluçon, France).

Cytotoxicity was monitored via the dosage of LDH released. LDH assays (Roche, Basel, Switzerland) were performed immediately before freezing. All cell experiments were done in triplicates.

### 4.4. Bacterial AHL Receptor Reporter Assay

The AHL receptor reporter strain contains a pSB1075 plasmid with genes encoding for tetracycline resistance and expression of the *Pseudomonas aeruginosa* AHL receptor LasR, as well as a fusion gene from the LasR promotor and the luxCDABE gene from *Photorhabdus luminescens*. On day 1, a 1/100 dilution of the bacterial strain (P1) was grown for 24 h at 37 °C under agitation in LB medium containing tetracycline. The bacterial culture was diluted on day 2, and in a black 96-well plate were placed 200 µL of P2 and 10 µL of compound (medium, water, and DMSO for negative controls, C4-HSL and 3-oxo-C12-HSL for positive controls, or tested sample). The plate was incubated for 4 h at 37 °C under agitation. Luminescence was read at endpoint at all wavelengths on a microplate reader. Data were normalized using the 3-oxo-C12-HSL as a 100%-induction response as a means for comparison.

### 4.5. Chemical Synthetic Procedures

Unless otherwise stated, all reactions were performed under argon atmosphere in dry glassware.

#### 4.5.1. Total Synthesis of 3-Oxo-substituted Natural AHL and Analogues 

The appropriate carboxylic acid was conjugated to Meldrum’s acid by EDC- and DMAP-mediated coupling. The resulting Meldrum’s acid derivative 5a-c underwent methanolysis by reflux in excess methanol. The ketone of the resulting 3-ketomethyl ester 6a-c was then ketal-protected with ethylene glycol in toluene, to afford the ketal-protected methyl ester 7a-c. The methyl ester was then removed through basic hydrolysis using NaOH in THF, to afford the free carboxylic acid 8a-c. The resulting ketal-protected carboxylic acid was linked to the appropriate headgroup, by means of EDC- and DMAP-mediated coupling in DCM. The ketone of the final intermediates 9–25 was freed by ketal-removal in TFA and water, to afford the desired final products 26–41. If necessary, at any stage of this synthesis, the intermediate was purified by flash chromatography [22,24,25,26,27,47,52,53].

#### 4.5.2. Synthesis of Terminal Azido-Carboxylic Acids 

The appropriate bromo-carboxylic acid (1.0 equiv) and sodium azide (1.5 equiv) were dissolved in DMF. The reaction was heated to 60 °C and stirred overnight under an argon atmosphere. At completion, the solvent was removed under reduced pressure. The resulting oil was dissolved in a 1:1:1 (*v*/*v*/*v*) mixture of EtOAc, H_2_O, and brine. Organic compounds were extracted with EtOAc and washed with half-saturated brine before drying (magnesium sulfate). Solvents were removed under vacuo to afford product as an oil [54].

#### 4.5.3. Synthesis of Non-3-Substituted AHLs 

To a stirred solution of (*S*)-(−)-(α)-amino-butyrolactone hydrochloride (1.0 equiv) in DCM were added Et3N (2.2 equiv) and the appropriate acyl chloride (1.0 equiv), on an ice bath. After dissolution, the ice bath was removed, and the mixture allowed to rise to room temperature. It was then stirred overnight at room temperature. At completion, the reaction mixture was washed with 1M HCl and brine, dried, and solvents were removed under vacuo to afford the desired product. If necessary, the product was purified by flash chromatography in EtOAc/cyclohexane [55].

#### 4.5.4. Synthesis of 3-Acyltetramic Acids from 3-Oxo-C12-HSL 

A solution of 0.5 M sodium methoxide in methanol (1.0 equiv) was added to a stirred solution of 3-oxo-C12-HSL (1.0 equiv) in methanol. The reaction mixture was stirred for 3 h at 55 °C and then overnight at 50 °C. The mixture was cooled to room temperature and then passed through an acidic ion exchange resin. The resin was eluted with MeOH, the eluents combined, and concentrated in vacuo to afford the desired tetramic acid as a solid. If necessary, the product was purified by flash chromatography [56,57].

### 4.6. Mass Spectrometry: Tandem LC/MS-MS

Evaluation of AHL half-life in cell culture medium at room temperature: The AHL was dissolved at 15 µM in DMEM GlutaMax and the solution stirred at room temperature. Samples (1 mL) were taken at intervals over 24 h, and extracted twice with 1 mL EtOAc, then acidified with 1M HCl solution to decrease pH to approximately 4–5, and extracted again with 1 mL EtOAc. All aqueous and organics phases were stored at −80 °C before treatment and analysis. For MS sample preparation, the organic extracts and aqueous phases were evaporated under a nitrogen stream at 50 °C. The residues were dissolved in 250 µL MeOH (HPLC grade) and placed in 2 mL MS vials (Agilent Technologies, Santa Clara, CA, USA) equipped with 250 µL inserts (Agilent Technologies, Santa Clara, CA, USA).

### 4.7. Kinetic Study of AHL Cellular Entry

Eighteen-day-old serum-starved Caco-2/TC7 were incubated with 1 mL/well of starvation medium containing 15 µM 3-oxo-C12-HSL, with or without 100 µM 2-HQ. At regular intervals, supernatants of selected wells were removed and placed in 1.5 mL Eppendorf microtubes, while the cells were scrapped in 500 µL ice-cold PBS 1X and lysed by sonication. All samples were extracted twice with EtOAc using a vortex and 3 min centrifugation at 5000 rpm. Aqueous phases were acidified with 20 µL of 1M HCl solution to bring the pH to 4, and extracted with EtOAc again following the same protocol. All aqueous and organics phases were stored at −80 °C before treatment and analysis. MS sample preparation was achieved as previously described.

### 4.8. AHL Detection by Tandem HPLC-MS/MS

The chromatographic separation of compounds was carried out on a Zorbax eclipse XDB-C18 column (Agilent Technology, Santa Clara, CA, USA) fitted on an Agilent 1100 tandem HPLC-MS/MS system. Column’s temperature was 45 °C. Then, 5 µL of compound solution were injected. The mobile phases consisted of (A) water with 0.1% formic acid and (B) acetonitrile with 0.1% formic acid in an 80:20 starting ratio, respectively. The linear gradient for AHL elution was programmed as follows: After increasing B in A from 20% to 40% over 5 min, and from 40% to 95% over 15 min, the gradient was kept constant over 10 min before 5 min re-equilibration (overall run time 35 min). Separation was achieved at a flow rate of 0.4 mL/min. This method allowed for separation of AHLs based on hydrophobicity.

Mass spectra were obtained using an API^®^ 2000 Q-Trap (AB-Sciex, Villebon sur Yvette, France) equipped with a TurboIon electrospray set in the positive mode with nitrogen as the nebulizer gas. The ion source temperature was set at 350 °C. Declustering and entrance potentials were set at 60 V and 5000 V, respectively. Data were acquired by the Analyst^®^ software (version 1.4.2, AB-Sciex, Villebon sur Yvette, France) in the multiple reaction monitoring mode.

AHL quantification was expressed in µmol/L after calibration with commercially available 3-oxo-C12-HSL and normalization relative to the internal standard *N*-butyryl-L-homoserinelactone-d5 (C4-*d5*-HSL, Cayman Chemicals, Ann Arbor, MI, USA).

### 4.9. Statistical Analysis

All data are represented as mean ± SEM of independent experiments and were tested for gaussian distribution. Statistical significance was examined by means of one-way ANOVA associated with post-test. Differences were considered significant at *p* < 0.05. All statistical analyses were realized using software Prism 6.0, GraphPad.

## 5. Patents

A patent “Analogues of *N*-acyl-homoserine lactones and Pharmaceutical composition comprising them” PAT2602795PC00 has been registered in France and submitted to the European Office of La Haye PCT/EP2020/079913 (23 October 2020).

## Figures and Tables

**Figure 1 ijms-21-09448-f001:**
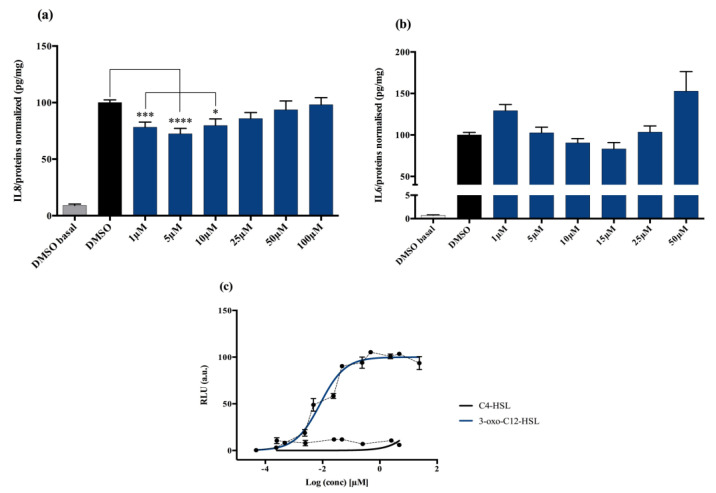
Cell response to 3-oxo-C12-HSL exposure: (**a**) IL-8 secretion of Caco-2/TC7 cells exposed to IL-1ß and a range of *N*-acyl-homoserine lactones (AHL) concentrations for 18 h. Secreted IL-8 is normalized to control DMSO. Points are the mean of several replicates (*n* > 10) ± SEM. * *p* < 0.05, *** *p* < 0.001, **** *p* < 0.0001 vs. control, one-way ANOVA (*p* < 0.0001), Tukey’s post-test. (**b**) IL-6 secretion of RAW264.7 cells exposed to LPS/TNF-α and a range of AHL concentrations for 6 h. Secreted IL-6 is normalized to control DMSO. Points are the mean of several replicates. (**c**) Bioluminescence of reporter strain *E. coli* pSB1075 with increasing doses of 3-oxo-C12-HSL and C4-HSL. Points are the mean value of several replicates (*n* > 3) ± SEM.

**Figure 2 ijms-21-09448-f002:**
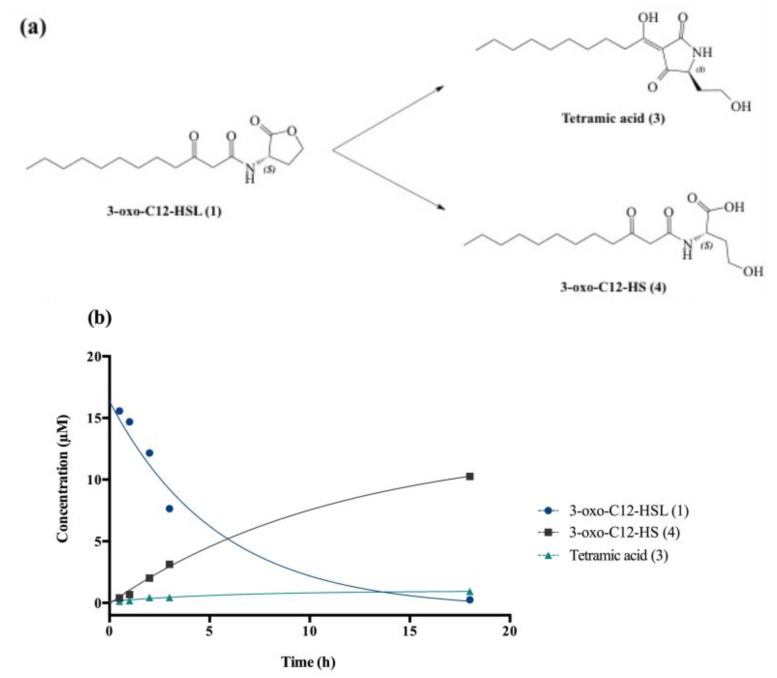
Degradation of 3-oxo-C12-HSL: (**a**) Degradation scheme into by-products (3) and (4). (**b**) Kinetics of decay of (1) in cell-free aqueous medium and apparition of tetramic acid (3) and 3-oxo-C12-HS (4).

**Figure 3 ijms-21-09448-f003:**
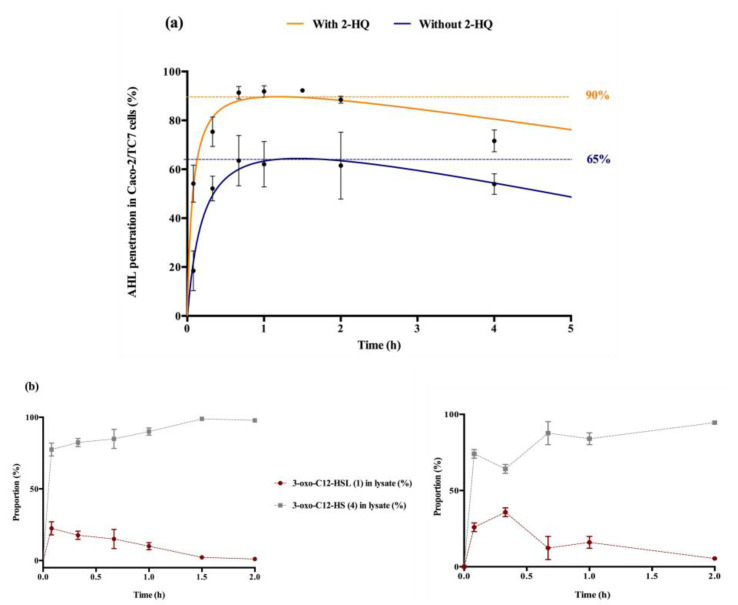
Cellular traffic of 3-oxo-C12-HSL: (**a**) Influence of 2-HQ on AHL penetration in Caco-2/TC7 cells over time, expressed as % of initial AHL concentration. (**b**) Time evolution of 3-oxo-C12-HSL and 3-oxo-C12-HS % in Caco-2/TC7 cell lysates in the presence (left) or absence (right) of 100 µM 2-HQ (truncated time profile). Points are the mean value of replicates (*n* = 3) ± SEM.

**Figure 4 ijms-21-09448-f004:**
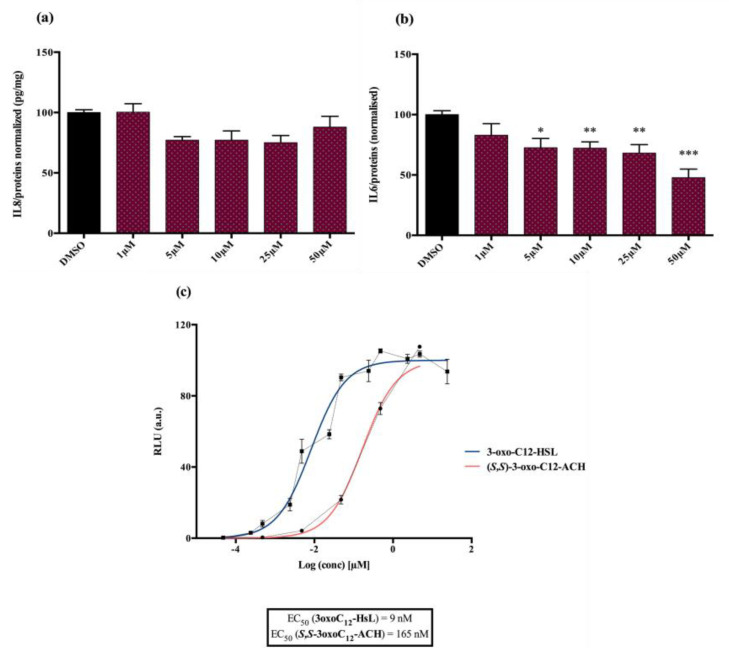
Biological activity of (*S*,*S*)-3-oxo-C12-ACH: (**a**) IL-8 secretion of Caco-2/TC7 cells exposed to inflammation and increasing doses of (*S*,*S*)-3-oxo-C12-ACH for 18 h. Secreted IL-8 normalized to control DMSO. Points are the mean of several replicates (*n* ≥ 8) ± SEM. No statistical difference vs. control observed in Tukey’s post-test, yet positive one-way ANOVA (*p* < 0.0001). (**b**) IL-6 secretion of RAW264.7 cells exposed to LPS/TNF-α and increasing doses of (*S*,*S*)-3-oxo-C12-ACH for 6 h. Secreted IL-6 normalized to control DMSO. Points are the mean of several replicates (*n* ≥ 8) ± SEM. * *p* < 0.05, ** *p* > 0.01, *** *p* < 0.001 vs. control, one-way ANOVA (*p* < 0.0001), Tukey’s post-test. (**c**) Compared bioluminescence in reporter strain *E. coli* pSB1075 with increasing doses of 3-oxo-C12-HSL (1) and (*S*,*S*)-3-oxo-C12-ACH (30). Points are the mean value of several replicates (*n* ≥ 6) ± SEM.

**Figure 5 ijms-21-09448-f005:**
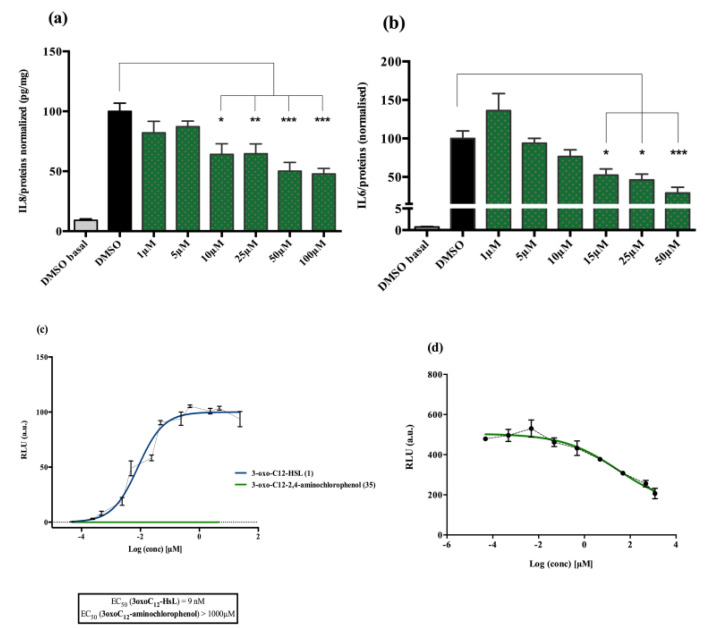
Biological activity of 3-oxo-C12-2-amino-4-chlorophenol: (**a**) IL-8 secretion of Caco-2/TC7 cells exposed to inflammation and increasing doses of 3-oxo-C12-2-amino-4-chlorophenol for 18 h. Secreted IL-8 normalized to control DMSO. Points are the mean of several replicates (*n* > 7) ± SEM. * *p* < 0.05, ** *p* > 0.01, *** *p* < 0.001 vs. control, one-way ANOVA (*p* < 0.0001), Tukey’s post-test. (**b**) IL-6 secretion of RAW264.7 cells exposed to LPS/TNF-α and increasing doses of 3-oxo-C12-2-amino-4-chlorophenol for 6 h. Secreted IL-6 normalized to control DMSO. Points are the mean of several replicates (*n* ≥ 8) ± SEM. * *p* < 0.05, *** *p* < 0.001 vs. control, one-way ANOVA (*p* < 0.0001), Tukey’s post-test. (**c**) Compared bioluminescence in reporter strain *E. coli* pSB1075 with increasing doses of 3-oxo-C12-HSL (1) and 3-oxo-C12-2-amino-4-chlorophenol (35). Points are the mean value of several replicates (*n* ≥ 6) ± SEM. (**d**) Inhibition of 3-oxo-C12-HSL-induced bioluminescence in *E. coli* pSB1075 with increasing doses of 3-oxo-C12-2-amino-4-chlorophenol. Points are the mean value of several replicates (*n* ≥ 6) ± SEM.

**Table 1 ijms-21-09448-t001:** Overview of the AHL analogues library.

*Entry*	*Common Name*	*Structure*
1	(*S*)-3-oxo-C12-HSL	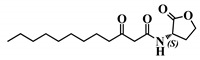
2	(*S*)-3-oxo-C12:2-HSL	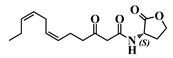
9	(*S*)-3-dioxolane-C12-HSL	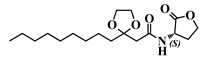
26	(*S*)-3-oxo-C6-HSL	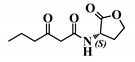
27	12-N3-(*S*)-3-oxo-C12-HSL	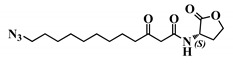
28	(*R*)-3-oxo-C12-HSL	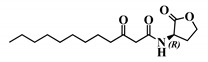
29	(*S*)-3-oxo-C12-HTL	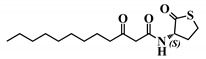
30	(*S*,*S*)-3-oxo-C12-ACH	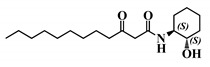
31	(*R*,*S*)-3-oxo-C12-ACH	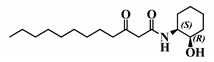
32	3-oxo-C12-*p*-methoxyanilide	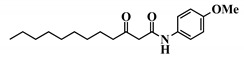
33	3-oxo-C12-o-methoxyanilide	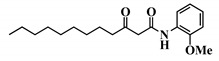
34	3-oxo-C12-*m*-methoxyanilide	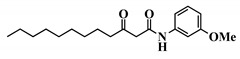
35	3-oxo-C12-2,4-aminochlorophenol	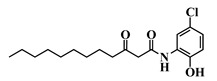
36	3-oxo-C12-Bz	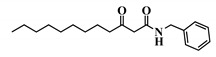
37	3-oxo-C12-aminopiperidine	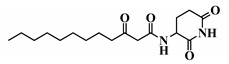
38	3-oxo-C12-fructose	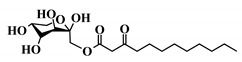
39	3-oxo-C12-Ala	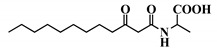
40	3-oxo-C12-ß-Ala	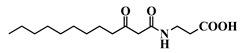
41	3-oxo-C12-Ser	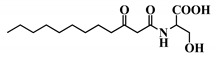
42	(*S*)-C4-HSL	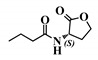
43	(*S*)-C12-HSL	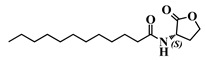
44	(*S*)-3-oxo-C14-HSL	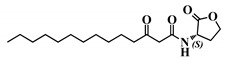

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
