# Peer review of "Anti-Inflammatory Effects of Analogues of N-Acyl Homoserine Lactones on Eukaryotic Cells"

_ijms, 2020, doi:10.3390/ijms21249448_

Round 1

Reviewer 1 Report

The authors analysed anti-inflammatory effects of 3-oxo-C12-HSL and analogues on epithelial and macrophage cell lines.

The manuscript is clearly written and experiments are straight forward.

The concentrations of the in vitro experiment are rather high. What would be the physiological range of 3-oxo-C12-HSL. The authors should comment on this and include it into the discussion.

Author Response

The authors are very thankful for the comments from reviewer#1. Please find some lines to address the question about physiological concentration of AHLs

It indeed seems quite important to examine the dosage of AHL employed in our experiments and its biological relevance. AHL have been identified in vivo in an array of fluids, from human sputum to bacterial biofilms, and now in faeces as reported by our team (Landman et al).

“Undirect” sampling – where samples are not taken in the immediate cellular surrounding - shows very low AHL concentrations. It is however probable that this method may only provide insights of the actual AHL levels or may undergo technical bias. Thus, concentrations ranging from 1 to 20 nM were detected in the sputum of patients with cystic fibrosis (Erickson et al.), while 0.2 to 2 nM/g of faeces were found in samples from IBD patients. On the opposite, very high concentrations are measured in Pseudomonas aeruginosa biofilms: levels as high as 300-600µM have been reported by Charlton et al, and biofilms have been identified in the lung of infected cystic fibrosis patients. Yet, our work aimed at exploring AHLs effects on cells lines in the context of gut inflammation driven by local dysbiotic microbiota. In this regard, it has been described that planktonic P. aeruginosa secreted AHLs at low micromolar concentrations (Pearson et al.), which is incidentally the range of concentration detected in a murine model of P. aeruginosa acute lung infection (1-20µM). Hence, we thought relevant to broaden this concentration range to 1-100µM, which could be the AHL dose range that intestinal cells from a IBD patient would encounter in a non-infected but inflammatory and dysbiotic state.

Erickson, D. L. et al. Pseudomonas aeruginosa quorum-sensing systems may control virulence factor expression in the lungs of patients with cystic fibrosis. Infect. Immun. 70, 1783–1790 (2002).

Charlton, T. S. et al. A novel and sensitive method for the quantification of N-3-oxoacyl homoserine lactones using gas chromatography-mass spectrometry: application to a model bacterial biofilm. Environ. Microbiol. 2, 530–541 (2000).

Pearson, J. P. et al. Structure of the autoinducer required for expression of Pseudomonas aeruginosa virulence genes. Proc. Natl. Acad. Sci. 91, 197–201 (1994).

Reviewer 2 Report

The manuscript titled " Anti-inflammatory Effects of N-Acyl Homoserine
Lactones Analogues on Eukaryotic Cells" is well designed, detailed study on the 3-oxo-C12 and their analogue derivatives of effects as anti-inflammatory property on the cell lines Caco-2 and RAW264.7.  It's interesting the analogues showed anti-inflammatory property without activating LasR receptor in the cell lines they tested. The conclusion were supported by results, it will be interesting study for the wider scientific community.  Below is the minor concerns in the manuscript, recommend for publication.  

i) Line 178-179 :  analogue (S,S)-3-oxo-C12-ACH showed strong effect on RAW264.7  than Caco-2 cells, could authors explain further reason for the difference in effect between these 2 cell lines?

Author Response

The authors are thankful for the comments from reviewer#2.

Epithelial cell lines and macrophages cells are quite different in terms of receptors and inflammatory pathways. For example, one of the most striking difference is their ability to response to LPS. Caco-2 epithelial cell line remains unresponsive to LPS as it does not exhibit activated TLR4-MD2 complex receptor. These cells can activate few inflammatory pathways and mainly IL-8 pathway to attract immune specialized cell on the site of inflammation. By contrast, macrophages such as RAW264.7 do express TLR4-MD2 complex and are highly responsive to LPS. Moreover, RAW264.7 are innate immune cells specialized in inflammatory responses to pathogens and/or changes in the micro-environment and can induce many cytokines and chemokines secretions pathways. Indeed, it is not surprising that those cells lines that represent together first line of defense in the gut can have different responses to molecules such as (S,S)-3-oxo-C12-ACH. Studying these responses from these 2 cells types in the context of gut inflammation provide relevant information on the local physiological responses.